# Reproducibility of Electromagnetic Field Simulations of Local Radiofrequency Transmit Elements Tailored for 7 T MRI

**DOI:** 10.3390/s25061867

**Published:** 2025-03-17

**Authors:** Max Joris Hubmann, Bilguun Nurzed, Sam-Luca Hansen, Robert Kowal, Natalie Schön, Daniel Wenz, Nandita Saha, Max Lutz, Thomas M. Fiedler, Stephan Orzada, Lukas Winter, Boris Keil, Holger Maune, Oliver Speck, Thoralf Niendorf

**Affiliations:** 1Faculty of Electrical Engineering and Information Technology, Otto von Guericke University, 39106 Magdeburg, Germany; joris.hubmann@ovgu.de (M.J.H.); robert.kowal@ovgu.de (R.K.); holger.maune@ovgu.de (H.M.); oliver.speck@ovgu.de (O.S.); 2Max-Delbrück-Center for Molecular Medicine in the Helmholtz Association (MDC), Berlin Ultrahigh Field Facility (B.U.F.F.), 13125 Berlin, Germany; bilguun.nurzed@mdc-berlin.de (B.N.); nandita.saha@mdc-berlin.de (N.S.); 3Chair of Medical Engineering, Faculty V, Technische Universität Berlin, 10587 Berlin, Germany; 4Faculty II, Berliner Hochschule für Technik, 13353 Berlin, Germany; 5Institute of Medical Physics and Radiation Protection, TH-Mittelhessen University of Applied Sciences, 35390 Gießen, Germany; sam-luca.hansen@lse.thm.de (S.-L.H.); boris.keil@lse.thm.de (B.K.); 6Research Campus STIMULATE, 39106 Magdeburg, Germany; 7Physikalisch-Technische Bundesanstalt (PTB), 10587 Braunschweig and Berlin, Germany; natalie.schoen@ptb.de (N.S.); max.lutz@ptb.de (M.L.); lukas.winter@ptb.de (L.W.); 8CIBM Center for Biomedical Imaging, 1015 Lausanne, Switzerland; daniel.wenz@epfl.ch; 9Animal Imaging and Technology, EPFL Swiss Federal Institute of Technology, 1015 Lausanne, Switzerland; 10Experimental and Clinical Research Center (ECRC), Charité—Universitätsmedizin Berlin, A Joint Cooperation Between the Charité Medical Faculty and The Max-Delbrück Center for Molecular Medicine in the Helmholtz Association, 13125 Berlin, Germany; 11German Cancer Research Center (DKFZ), 69120 Heidelberg, Germany; t.fiedler@dkfz-heidelberg.de (T.M.F.); stephan.orzada@dkfz-heidelberg.de (S.O.); 12LOEWE Research Cluster for Advanced Medical Physics in Imaging and Therapy (ADMIT), TH-Mittelhessen University of Applied Sciences, 35390 Giessen, Germany; 13Department of Diagnostic and Interventional Radiology, University Hospital Marburg, Philipps University of Marburg, 35043 Marburg, Germany; 14Faculty of Natural Sciences, Otto von Guericke University, 39106 Magdeburg, Germany

**Keywords:** numerical simulations, reproducibility, radiofrequency, RF transmit elements, MRI, ultrahigh-field MRI, validation

## Abstract

The literature reports on radiofrequency (RF) transmit (Tx) elements tailored for ultrahigh-field (UHF) magnetic resonance imaging (MRI) showed confounded reproducibility due to variations in simulation tools, modeling assumptions, and meshing techniques. This study proposes a standardized methodology to improve reproducibility and consistency across research sites (testers) and simulation tools (testing conditions). The methodology includes detailed simulation workflow and performance metrics for RF Tx elements. The impact of the used mesh setting is assessed. Following the methodology, a reproducibility study was conducted using CST Microwave Studio Suite, HFSS, and Sim4Life. The methodology and simulations were ultimately validated through 7 T MRI phantom experiments. The reproducibility study showed consistent performance with less than 6% standard deviation for *B*_1_^+^ fields and 12% for peak SAR averaged over 10 g tissue (pSAR_10g_). The SAR efficiency metric (|*B*_1_^+^|/√pSAR_10g_) was particularly robust (<5%). The simulated and experimental |*B*_1_^+^| maps showed good qualitative agreement. This study demonstrates the feasibility of a standardized methodology for achieving reproducible RF Tx element electromagnetic field simulations. By following the FAIR principles including making the framework publicly available, we promote transparency and collaboration within the MRI community, supporting the advancement of technological innovation and improving patient safety in UHF-MRI.

## 1. Introduction

Magnetic resonance imaging (MRI) at 1.5 T and 3 T is a cornerstone of medical diagnostic imaging. Ultrahigh-field MRI (*B*_0_ ≥ 7 T, UHF-MRI) provides groundbreaking opportunities for advancing biomedical and diagnostic MRI. UHF-MRI revealed more anatomical detail and pathophysiological characteristics due to enhanced sensitivity and spatial resolution [1,2,3,4,5,6]. This gain has served as a driving force for numerous applications in basic research and clinical science resulting in a wide range of new possibilities [1,2,3,4,5,6,7,8,9,10,11,12]. However, the opportunities of UHF-MRI are obstructed by electromagnetic field constraints including transmission field (*B*_1_^+^) inhomogeneities due to wavelength (*λ*) shortening and radiofrequency (RF) power deposition constraints [12,13].

These obstructions constrain the broader clinical application of UHF-MRI and have motivated the exploration of novel RF technologies, including local transceiver (Tx/Rx) arrays and multi-channel transmission (Tx) arrays, paired with multi-channel local receive (Rx) arrays. The RF building blocks to assemble Tx arrays can be loop, stripline, and dipole element designs including hybrid loop–dipole configurations [14,15,16,17,18,19]. The Tx array designs involve rigid, flexible, and lightweight configurations and are tailored to conform to the target anatomy and to accommodate multiple body habitus and anatomical variants [16,19,20,21,22,23,24]. The number of RF building blocks and their density and positioning influence the Tx array design to ensure an excitation profile best suited for covering the target region [20,23,25,26,27]. This has led to a very diverse spectrum of Tx array configurations customized for specific applications virtually ranging from head to feet [16,19,20,21,22,23,24]. While the customization and diversity of Tx array designs are beneficial, they constitute challenges for validation, performance assessment, and quality assurance. Furthermore, the benchmarking of Tx elements poses a serious challenge due to the variety of specific setups and applications involved. A real-world example is provided in Table 1, which summarizes the Tx performance reported for eight Tx element configurations customized for body UHF-MRI. These valuable reports underline the incoherence and diversity of the setups and metrics used for the assessment of Tx elements.

A critical aspect of advancing RF antenna designs is the need for reproducibility and standardization in evaluation protocols. Evaluation protocols for self-developed RF arrays typically describe internal procedures, tier-based formalities, and safety testing recommendations [34,35,36]. While these efforts have contributed to the field, they lack wide dissemination, fail to represent a broad consensus, and do not consistently apply the Findable, Accessible, Interoperable, Reproducible (FAIR) principles. Reproducibility refers to the ability to obtain consistent results under varying test conditions and testers [37]. However, reproducibility remains a significant challenge, especially for in silico modeling. In silico modeling has become a powerful tool for RF technology developments and offers substantial benefits regarding resources, time, and safety. Yet variations in software, modeling assumptions, material parameters, and many more parameters often lead to discrepancies between studies, making it difficult to achieve consistent results across different modeling platforms and research groups. For optimal outcomes, in silico modeling should be performed with standardized modeling protocols to improve reproducibility, ensuring real-world applicability, safety, and reliability, and complemented with experimental validation. Moreover, to date, there are no clear guidelines outlining which simulation parameters should be included in scientific or engineering reports to ensure reproducibility across different sites and tools used for electromagnetic field (EMF) simulations. This absence has been a significant obstacle to technology transfer, reproducibility, and the clinical translation of UHF-MRI [37,38]. Therefore, standardized methodologies are essential to enhance the reproducibility of RF technology [39]. Recognizing the challenges and opportunities, this work applies FAIR principles and proposes an exemplary methodology for the reproducible assessment of local Tx elements (FAI**R**) tailored for UHF-MRI. It also suggests an exemplary guideline on how to report novel Tx element designs. The guideline includes well-defined methodology and performance metrics for a certain use case to guarantee good reproducibility (FAI**R**). For this purpose, four Tx elements developed for UHF-MRI were reproduced and assessed following the proposed guidelines. This reproducibility study was conducted using five EMF simulation solvers across three imaging sites (FA**IR**). By sharing our guidelines and protocol in an open-access online repository (**FA**IR), we aim to promote technology transfer and clinical translation, emphasizing reproducibility. This will help lower barriers to the assessment of RF coil technology, benefitting users and MRI engineers with all levels of experience.

## 2. Materials and Methods

To enhance and standardize the assessment of local Tx elements, this study provides a guideline on the parameters essential to guarantee the reproducibility of EMF simulations of Tx elements for UHF-MRI under varying testing conditions and testers. The values provided should be viewed as flexible guidelines rather than a strict framework, as they may vary depending on the specific use case of the Tx element concept. In this work, a meshing technique was first determined based on mesh analysis to perform EMF simulations on four Tx elements reported in the literature. Detailed descriptions of the performance assessment methodology are provided in Section 2.1. This exemplary methodology was then applied to reproduce the results, which were ultimately validated through 7 T MRI measurements.

### 2.1. EMF Simulation Setup and Workflow

The design and performance assessment of Tx elements for UHF-MRI EMF simulations in the literature was conducted using commercially available or open-source software tools. The used software tool and version should be reported due to different settings of the software tool. In this study, the commercially available software CST Studio Suite 2020 and 2022 (Dassault Systèmes, Vélizy-Villacoublay Cedex, France), Sim4Life Version 8.0.1.15737 (Zurich Med Tech AG, Zurich, Switzerland), HFSS Version 2021R1 (ANSYS, Inc., Canonsburg, PA, USA) was used.

EMF simulations were performed in the high-frequency time and frequency domain solver at a frequency of 297.2 ± 50 MHz (7 T). The high-frequency time domain solver is based on the finite integration technique (FIT) for the CST simulations and on the finite differential time domain (FDTD) method for the simulations in Sim4Life. The frequency domain solver of CST and HFSS are based on the finite element method (FEM). The solvers offer different settings as well as meshing techniques, which play a vital role [40,41] and, therefore, need to be reported.

In the presented simulations, a homogenous phantom with specific dimensions and dielectric parameters mimicking a certain part of the body was modeled, and usually, these parameters are reported as given in Table 1. Additionally, components of the MRI system such as the bore can be modeled. In this work, a rectangular, uniform phantom (Figure 1) (x = 300 mm, y = 150 mm z = 400 mm; *ε*_r_ = 59, *σ* = 0.83 S/m, *ρ* = 1058 kg/m^3^ at 297.2 MHz) mimicking average body tissue and the upper torso size was used without the MRI bore. The housing of the phantom was modeled with 8 mm PMMA (due to a lack of reliable literature values using *ε*_r_ = 3.4, *σ* = 0.0043 S/m, *ρ* = 1180 kg/m^3^ at 297.2 MHz) and a 4 mm PMMA lid. This phantom configuration has low geometric complexity, which makes the reproducibility study as well as the validation less prone to errors resulting from geometric differences. The used material properties at the desired frequency should always be reported because default material values can vary between software tools.

The Tx elements were modeled following the design parameters outlined in the original literature reports or were provided by the original researcher. For better technology transfer and reproducibility, the Tx element models should be described in detail, with dimensions, material parameters, and used lumped elements, and/or should be provided by an open-access online repository. In this study, four Tx elements were modeled and assessed (Figure 2): (1) rectangular loop with distributed capacitors (LP, 105 × 55 mm^2^), (2) fractionated dipole with inductors as meander (FD, Inductors = 33.5 nH) [30], (3) microstrip with meander (MS, Capacitors = 1 pF) [28], and (4) snake antenna (SN) [32]. The tuning and matching networks should also be reported. The tuning and matching networks were modeled in the post-processing according to Figure 2. Here, the Tx elements were tuned and matched to |*S*_11_| < −50 dB by adjusting the lumped elements values of the tuning and matching network. In the Appendix A, a description of the post-processing for tuning and matching is provided. The conductors were modeled as perfect electric conductors (PECs) with infinitesimally small thickness and FR-4 substrate was used (*ε*_r_ = 4.3, loss tangent = 0.012 @297.2 MHz).

Certain Tx element concepts like loops or the fractionated dipole require lumped elements. Usually, the capacitor or inductor values are reported but not always the considered losses. The used lumped elements together with the losses should be reported and/or provided by the online repository. In this study, the losses were estimated with an Equivalent Series Resistance (ESR) for capacitors and quality (Q)-factors for inductors. The ESR and Q-factor values were taken from ATC Tech-SELECT 9.0 (Scillasoft Consulting, Bellefonte, PA, USA). The losses can be determined from different data sheets but need to be reported because of the impact on the power and field distribution. The Tx element designs used in this study are shown in Figure 2 and provided by the online repository (see Data Availability).

The Tx elements were placed concentric with a 20 mm distance to the phantom. The distance has a huge impact on the transmit performance and, therefore, should always be reported, which is not always the case as shown in Table 1. On the one hand, the distance changes the loading of the Tx elements, and on the other hand, more/less energy is transferred into the target object.

Once the Tx element is modeled on the phantom, a mesh setup needs to be chosen. The quality and type of the meshing technique plays a critical role in EMF simulations [40,41] and, therefore, needs to be reported so that other users with different software tools can reproduce them. In this study, a CST-specific local mesh refinement of the components was performed as outlined in Section 2.2. The hexahedral mesh cells can be set as absolutes in mm or relative to the size of the object. The presented local mesh refinement should be seen as a minimum criterion and should not fall below the number of mesh cells or above the absolute mesh cell size. The substrate (FR-4) of the Tx elements was meshed with at least 2 mesh cells relative to the thickness (e.g., 2 mm thick substrate needs at least 1 mm mesh cells). The traces (PEC) of the Tx element were also meshed with at least 2 mesh cells relative to the width (e.g., 5 mm width:2.5 mm mesh cells). Gaps for ports and lumped elements were also meshed with at least 2 mesh cells relative to the gap size (e.g., 3 mm gap:1.5 mm mesh cells). The phantom was meshed with an absolute mesh size of 4 mm isotropic. To avoid standing wave effects, open boundary conditions were applied in all directions of the simulation environment using a perfectly matched layer. The air domain around the model was left on default. Detailed settings can be found in the Appendix A.

To benchmark Tx elements, standardized metrics should be used. As a starting point, two of the most common transmit performance parameters were assessed: (i) power efficiency (|*B*_1_^+^|/√*P*) and (ii) specific absorption rate (SAR) efficiency (|*B*_1_^+^|/√peak SAR averaged over 10 g (pSAR_10g_)) normalized to 1 W stimulated power at the Tx elements input. The pSAR_10g_ was evaluated following the IEC/IEEE 62704-1:2017 guideline [42]. For better reproducibility and better comparison, a quantitative value at a certain evaluation point should be used as a reference point and reported or highlighted in the plot. In this study, a center point on the phantom solution’s surface (at 0 mm depth) was used. The metrics of power efficiency and SAR efficiency were supplemented by two relevant performance metrics for in vivo applications. RF array performance is deteriorated by the intrinsic coupling of the individual Tx elements. Therefore, the (iii) intrinsic decoupling (|*S*_21_|) of two Tx elements was assessed with center-to-center distances of 100 mm between the Tx elements. This distance can vary depending on the array setup but needs to be reported. For in vivo applications, the (iv) loading of the Tx elements changes due to variations in the body size and placement of the Tx elements. The distance of the Tx elements is usually reproducible, but the variance between patients results in different loading conditions. Therefore, the loading dependence of the Tx elements is a critical performance criterion. To assess this criterion and to mimic different loading conditions, the permittivity and conductivity of the phantom solution were reduced by 30% without adjusting the tuning and matching network, whereby the remaining |*S*_11_| parameter at 297.2 MHz as well as the frequency change Δ*f* of the minimum |*S*_11_| parameter was assessed.

The resulting workflow and settings for reproducible Tx element EMF simulations are summed up in Figure 3.

### 2.2. Mesh Analysis

As mentioned before, the quality and type of the meshing technique play a critical role in EMF simulations [40,41,43]. It will affect the values of the tuning and matching network as well as the EMF results. Therefore, a mesh analysis in CST Studio Suite 2020 was performed using the four described Tx elements (Figure 2). The used meshing technique may vary between software tools and the versions; therefore, a CST-specific meshing technique was applied in this study. The Tx elements were placed on the reported rectangular phantom (Figure 1). The number of mesh cells, as well as the computational time, was determined with GPU acceleration (Nvidia Titan RTX 24 GB, CPU: AMD EPYC 7402 24-Core base 2.8 GHz base speed). The accuracy was assessed using the transmit efficiency (|*B*_1_^+^|/√*P*) normalized to 1 W of stimulated power at a depth of 0 mm (center axis) inside the phantom solution as well as the pSAR_10g_. For the mesh analysis, the accuracy of the following two high-frequency CST Studio Suite solvers for 3D full-wave simulations was compared:

(i) FEM of the frequency domain solver with automatic adaptive tetrahedral meshing as a baseline. The adaptive meshing technique eliminates the need for additional manual local mesh refinement or the specification of maximum/minimum mesh cell sizes. A threshold of 0.01 for the *S*-parameters was used as a termination criterion. The threshold represents the maximum acceptable deviation for the *S*-parameter change after each mesh refinement. The Tx elements were tuned and matched to |*S*_11_| < −50 dB.

(ii) FIT of the time domain solver with varying maximum hexahedral mesh cell size. The setting defines the largest allowed cell size in the simulation space in the absence of other local refinements, where the maximum cell is given by cells per wavelength (λ). A solver accuracy of −60 dB [41] was used as a termination criterion. In this case, the solver stops when the remaining energy in the calculation domain decreases to −60 dB compared to the maximum energy. The *cells per wavelength* were decreased from 30 *cells per wavelength* to 5 cells per wavelength (step width = 1 cell) and benchmarked against the FEM solver results. The highest frequency of interest determines the smallest wavelength. For each iteration, the Tx elements were tuned and matched to |*S*_11_| < −50 dB.

(iii) Furthermore, local mesh refinement using the FIT of the time domain solver was performed with 20 *cells per wavelength* as the largest allowed mesh cell size. For the Tx element, the substrate (FR-4) was meshed with 2 mesh cells relative to the thickness (e.g., 2 mm thick substrate needs 1 mm mesh cell). The traces (PEC) of the Tx element were also meshed with 2 mesh cells relative to the width (e.g., 5 mm width:2.5 mm mesh cell). Gaps for ports and lumped elements were meshed with 2 mesh cells relative to the gap size (e.g., 3 mm gap:1.5 mm mesh cell). The phantom was meshed with an absolute mesh size of 4 mm isotropic. The Tx elements were tuned and matched to |*S*_11_| < −50 dB and benchmarked against the FEM solver results.

### 2.3. Reproducibility

The reproducibility study applied the outlined methodology for EMF simulations under varying testing conditions and testers. Specifically, EMF simulations were conducted using five different solvers across four distinct simulation software tools. These tests were performed at three separate research sites, each representing different tester environments. The previously described Tx elements (Figure 2) were simulated in two CST versions (2020, 2022), where for 2020, the FEM and the FIT solver were used. These simulation tools apply different methods to calculate the field as well as different meshing techniques. CST uses the FEM with a tetrahedral mesh and FIT with a hexahedral mesh. Sim4life uses a hexahedral mesh applying a FDTD, and HFSS has a tetrahedral mesh while performing a FEM simulation. The models of the four Tx elements and the phantom were provided by the online repository, https://github.com/AntennaComp (accessed on 1 November 2024) (see Data Availability). The testers performed EMF simulations following the EMF simulation setup and workflow (Section 2) without any assistance. Tuning and matching were performed individually with the suggested network of the guideline if possible (|*S*_11_| ≤ −50 dB). The values of the lumped elements of the tuning and matching network are reported as well as the ESR and Q values. The described benchmarking metrics were assessed for each software solver (testing condition).

### 2.4. Validation

To validate the simulations, the described phantom was built using deionized water with polyvinylpyrrolidone (PVP) to achieve a relative permittivity of *ε*_r_ = 59. Sodium chloride (NaCl) was used to define the conductivity of *σ* = 0.83 S/m [44,45]. The solution was heated to 60 °C with constant stirring to produce a uniform solution. The concentration of PVP and NaCl was 371 g/L and 13.5 g/L of water, respectively. The permittivity and conductivity were measured with a dielectric probe (SPEAG’s Dielectric Assessment Kit DAK 12 Probe 4 MHz–3 GHz, Schmid & Partner Engineering AG, Zurich, Switzerland). The four Tx elements were placed on the phantom as described in the simulation setup (Figure 1) and connected to the MRI system via a T/R switch box (Stark Contrast, Erlangen, Germany). MRI experiments were conducted on a 7 T whole-body MR scanner (MAGNETOM 7 T, Siemens Healthineers AG, Erlangen, Germany). The Tx elements were tuned and matched to 297.2 MHz according to Figure 2. |*B*_1_^+^|-field measurements were conducted with a slab-selective actual flip angle imaging (AFI) method [46] (spatial resolution = 4 × 4 × 4 mm^3^, sinc pulse (pulse duration = 1 ms), TE = 2.19 ms, TR1 = 20 ms, TR2 = 100 ms, receiver bandwidth = 500 Hz/Px, nominal FA = 50°, V_ref_ = 80 V/100 V). |*B*_1_^+^|-maps were reconstructed offline in MATLAB 2021b (Mathworks, Natick, MA, USA). The AFI sequence provides precise |*B*_1_^+^|-field measurements between 20° and 70° of actual flip angles [46]. Losses in the signal chain of the MRI system between the RF amplifier and the feeding ports of the Tx element were measured with an 8-channel vector network analyzer (Rohde & Schwarz GmbH & Co. KG, Munich, Germany). The total losses within the signal cable and the T/R switch were 3.35 dB. Furthermore, the estimated material losses as well as measured input port reflections |*S*_11_| were considered in the simulation results for the validation. The measured |*B*_1_^+^|-maps were compared to the simulated |*B*_1_^+^|-maps.

## 3. Results

### 3.1. Mesh Analysis

The mesh analysis results of the FIT solver relative to the FEM solver are illustrated in Figure 4. Table 2 summarizes the results of the (i) FEM solver with adaptive mesh refinement with the used lumped elements to tune and match the Tx elements. In Table 3, the results of the (ii) FIT solver with 30 mesh cells per *λ* are shown together with the lumped elements. The MS could only be tuned and matched to |*S*_11_| < −13.7 dB. Under the same tuning and matching network, the frequency shifted with decreasing maximum mesh cell size (Figure 5). For the comparison of the decreasing number of mesh cells per *λ*, the Tx elements were then tuned and matched to |*S*_11_| < −50 dB. Only for the MS, the |*S*_11_| < −13.7 dB could not be improved. The results of the (ii) FIT solver with 5 mesh cells per *λ* are summarized in Table 4. The results of the (iii) FIT solver using 20 cells per *λ* with additional local mesh refinements are shown in Table 5. The power efficiency showed a worst-case deviation of up to −57% and a pSAR_10g_ of up to −33% with coarse meshing using the FIT solver relative to the FEM solver.

### 3.2. Reproducibility

The results of the reproducibility study are summarized in Table 6, Table 7, Table 8 and Table 9. For Sim4Life, the lumped elements of the tuning and matching network were modeled loss-free because the lumped elements could not be modeled with ESR or *Q*-values in the co-simulation.

The loop was tuned with capacitors (*C*_T_) ranging between 8.2 pF (CST 2020 FEM) and 8.6 pF (Sim4Life) and matched with capacitors (*C*_M_) between 16.6 pF (CST 2020 FEM) and 21.1 pF (HFSS). The mean ± standard deviation (SD) |*B*_1_^+^| of the LP was 1.79 ± 0.06 µT/√W. The highest |*B*_1_^+^| was obtained using Sim4Life and FEM solver of CST 2020 (1.84 µT/√W) and the lowest |*B*_1_^+^| using the FIT solver of CST 2020 (1.74 µT/√W). The mean ± SD (pSAR_10g_) was 1.79 ± 0.10 W/kg. The highest pSAR_10g_ was obtained using Sim4Life (1.89 W/kg) and the lowest using the FEM solver of CST 2020 (1.60 W/kg). The mean ± SD SAR efficiency was 1.34 ± 0.07 µT/√(W/kg). The highest SAR efficiency was obtained using the FEM solver of CST 2020 (1.45 µT/√(W/kg)) and the lowest using the FIT solver of CST (1.29 µT/√(W/kg)), which reveals a maximum deviation of 12.4%. For the coupling assessment, the LP showed mean ± SD coupling losses of 6.26 ± 0.71%. The highest coupling losses were obtained with the FIT solver in CST 2020 (6.76%) and the lowest for HFSS (5.01%). The loading dependence showed mean ± SD |*S*_11_| of 1.10 ± 0.14%. The mean ± SD frequency shift was 1.02 ± 0.11 MHz. The highest |*S*_11_| as well as frequency shift was obtained for HFSS (1.32%, Δ*f* = 1.2 MHz).

The FD was tuned and matched with inductors between 13.4 nH (HFSS) and 58.7 nH (Sim4Life) and with capacitors between 45.4 pF (FIT CST 2020) and 6.9 pF (HFSS). The mean ± SD |*B*_1_^+^| of the FD was 1.25 ± 0.03 µT/√W. The highest |*B*_1_^+^| was obtained using HFSS (1.30 µT/√W) and the lowest |*B*_1_^+^| using the Sim4Life (1.22 µT/√W). The mean ± SD (pSAR_10g_) was 1.41 ± 0.11 W/kg. The highest pSAR_10g_ was obtained using HFSS (1.56 W/kg) and the lowest using the FEM solver of CST 2020 (1.23 W/kg). The mean ± SD SAR efficiency was 1.05 ± 0.04 µT/√(W/kg). The highest SAR efficiency was obtained using the FEM solver of CST 2020 (1.12 µT/√(W/kg)) and the lowest using the FIT solver of CST and Sim4Life (1.03 µT/√(W/kg)), which reveals a maximum deviation of 8.7%. For the coupling assessment, the FD showed mean ± SD coupling losses of 2.44 ± 0.16%. The highest coupling losses were obtained with HFSS (2.69%) and the lowest for the FEM solver of CST 2020 (2.24%). The loading dependence showed mean ± SD |*S*_11_| of 0.49 ± 0.17%. The mean ± SD frequency shift was 1.58 ± 0.47 MHz. The highest |*S*_11_| (0.79%) and the highest frequency shift (Δ*f* = 2.4 MHz) were obtained using the FIT solver of CST 2022.

The two feeding ports of the MS with *λ*/2 phase shift transmission line could not be realized in Sim4Life. Each feeding port was tuned and matched with a serial and parallel capacitor (*C*_T_ = 9.2 pF, *C*_M_ = 3.9 pF) individually to −50 dB. Both ports were simulated with the same phase, and later in the post-processing, a λ/2 phase shift was applied. For the other simulation tools, the MS was tuned with capacitors between 4.8 pF (FEM CST) and 10.6 pF (HFSS) and matched with capacitors between 5.5 pF (FEM CST) and 6.8 pF (HFSS). The mean ± SD |*B*_1_^+^| of the MS was 1.45 ± 0.04 µT/√W. The highest |*B*_1_^+^| was obtained using Sim4Life (1.51 µT/√W) and the lowest |*B*_1_^+^| using the FEM solver of CST 2020 (1.43 µT/√W). The mean ± SD (pSAR_10g_) was 1.84 ± 0.07 W/kg. The highest pSAR_10g_ was obtained using Sim4Life (1.94 W/kg) and the lowest using the FEM solver of CST 2020 (1.76 W/kg). The mean ± SD SAR efficiency was 1.07 ± 0.01 µT/√(W/kg). The lowest SAR efficiency was obtained using the HFSS (1.05 µT/√(W/kg)), while the other simulation tools showed similar SAR efficiencies of around 1.08 µT/√(W/kg), which reveals a maximum deviation of 2.8%. For the coupling assessment, the MS showed mean ± SD coupling losses of 1.23 ± 0.31%. The highest coupling losses were obtained with the FIT solver of CST 2020 (1.78%) and the lowest for Sim4Life and CSTs FEM solver (1.07%). The loading dependence showed mean ± SD |*S*_11_| of 0.27 ± 0.11%. The mean ± SD frequency shift was 0.3 ± 0.1 MHz. The highest |*S*_11_| was obtained using FIT CST 2022 (0.43%), and the highest frequency shift was obtained for CST FIT (Δ*f* = 0.4 MHz).

The SN could not be tuned and matched with a lattice balun in Sim4Life. The tuning and matching network was modeled with a serial (*C*_S_) and parallel capacitor (*C*_P_). The *C*_p_ was 0.1 pF and the *C*_S_ was 2.1 pF. For the other simulation tools, the SN was tuned with a balun. The capacitors of the balun were between 3.66 pF and 4.18 pF for all simulation tools. The inductors of the balun were between 21.43 nH (FEM CST 2020) and 28.29 nH (FIT CST 2022). The mean ± SD |*B*_1_^+^| of the SN was 1.16 ± 0.02 µT/√W. The highest |*B*_1_^+^| was obtained using Sim4Life (1.20 µT/√W), while the other simulation tools showed a |*B*_1_^+^| of around 1.15 µT/√W. The mean ± SD (pSAR_10g_) was 1.46 ± 0.12 W/kg. The highest pSAR_10g_ was obtained using Sim4Life (1.56 W/kg) and the lowest using the FEM solver of CST 2020 (1.25 W/kg). The mean ± SD SAR efficiency was 0.97 ± 0.04 µT/√(W/kg). The highest SAR efficiency was obtained for the FEM solver of CST 2020 (1.03 µT/√(W/kg)) and the lowest using HFSS (0.94 µT/√(W/kg)), which reveals a maximum deviation of 8.7%. For the coupling assessment, the SN showed mean ± SD coupling losses of 3.67 ± 0.66%. The highest coupling losses were obtained with Sim4Life (4.68%) and the lowest for the FEM solver of CST 2020 (2.95%). The loading dependence showed mean ± SD |*S*_11_| of 0.46 ± 0.05%. The mean ± SD frequency shift was 0.6 ± 0.2 MHz. The highest |*S*_11_| was obtained using Sim4Life (0.51%), and the highest frequency shift was obtained for CSTs FIT solver (Δ*f* = 0.7 MHz).

### 3.3. Validation

For validation, simulated |*B*_1_^+^|-maps were benchmarked against |*B*_1_^+^| measurements. The measured and simulated axial |*B*_1_^+^|-maps obtained at the center of the phantom are depicted in Figure 6. The simulated and experimental |*B*_1_^+^|-maps matched qualitatively. The quantitative analysis revealed a relative difference between measurement and simulation ≤ 5% for the LP in the majority of the measurable area (flip angle between 20° and 70°) and only increases up to 10% on the left side in a circular area. For the FD and the SN, a difference of <5% was found for the center area, while it was slightly pronounced at the edges of the measurable region. The MS showed a difference between measurement and simulation of <10% for most of the area, whereby it was slightly pronounced in the center of the element. Generally, the simulations generated higher |*B*_1_^+^| values compared to the measurement.

## 4. Discussion

In this work, we proposed a standardized methodology for single Tx elements tailored for UHF-MRI that enhances the reproducibility of EMF simulations and applies the FAIR principles. For the first time, a reproducibility study involving four different Tx elements for 7 T MRI was conducted across different simulation tools and UHF sites. The reproducibility study showed consistent performance with SD of less than 6% for |*B*_1_^+^| fields and 12% for pSAR_10g_. The SAR efficiency metric (|*B*_1_^+^|/√pSAR_10g_) was particularly robust (<5%), with minimal variability across different sites. To improve reproducibility, ensuring real-world applicability, safety, and reliability, the proposed methodology was validated through 7 T MRI experiments. Qualitatively, the experimental data were in accordance with the simulation results, showing comparable Tx field patterns with a local maximum deviation of 10%.

In silico modeling has become a powerful tool for RF technology developments for UHF-MRI. The meshing technique for in silico modeling plays a pivotal role not only in accurately assessing the |*B*_1_^+^| field but also in ensuring patient safety [40,41,43]. SAR calculations are particularly sensitive to model resolution. Previous studies demonstrated a 56% variation [41] of SAR between 2.25 mm and 0.8 mm (isotropic) model resolution and a 70% variation [43] between 5 mm and 2 mm model resolution. Such discrepancies are primarily caused by coarse meshing, which impacts port impedance, lumped element properties, and the mass-averaging method [43]. The underestimation of SAR due to insufficient mesh resolution poses serious risks to patient safety, reinforcing the need for standardized and transparent reporting practices in RF modeling research. The optimal mesh setup with improved accuracy needs to be validated through E- and H-field measurements [47] as well as phantom measurements. The observed local discrepancy of 10% in the validation between |*B*_1_^+^| obtained from simulations and measurements is comparable with errors reported in the literature, ranging from 10% to 50% [16,30,31,48,49,50], and therefore, represents an appropriate simulation setup for the assessment of Tx elements within reasonable simulation times.

This study investigated the variations in |*B*_1_^+^| and pSAR_10g_ results across different simulation solvers, meshing techniques, and mesh resolutions for single Tx elements on a homogeneous phantom. The complexity of Tx element designs significantly impacted the accuracy of results. More complex designs required finer meshing, achieved either through the adaptive meshing technique of the FEM solver or enhanced mesh resolution with the FIT solver. This can be seen by the S-matrix (Figure 4) of the Tx elements, where simple designs such as the LP or FD show a resonance frequency at 297.2 MHz, and with varying meshing resolution, a shift could be obtained. The complex designs of the MS and SN showed either insufficient tuning and matching (|*S*_11_| > −13.7 dB) or no change with varying mesh resolution. This behavior is due to the different meshing resolutions affecting the electrical length of the Tx element, hence the port impedance. The meshing technique and the false calculation of the port impedance affected the lumped element properties required to tune and match the Tx element. The lumped element values of the (i) FEM solver were comparable with the values of the (iii) FIT solver using additional local mesh refinements (Table 2 and Table 5). Using coarse meshing of the (ii) FIT solver resulted in a deviation of the lumped element values.

The insufficient meshing and false calculation of the port impedance revealed for the |*B*_1_^+^| fields an underestimate by up to −57% in this study. This was particularly notable when using the FIT solver compared to the FEM solver. This underestimation may primarily be attributed to the FEM solver’s adaptive meshing, which ensures finer meshing of antenna elements. Increasing the number of mesh cells per wavelength in the FIT solver compensated for this underestimation. Adding local mesh refinements in the FIT solver further enhanced accuracy, and by considering this, the ground truth in some cases outperformed the FEM solver. The overestimation in pSAR_10g_ for the LP and the FD of the FEM solvers was largely due to the adaptive meshing, which resulted in fine meshing of conducting materials while applying coarser meshing to the phantom. For more complex designs, such as the MS with meander structures or the SN with curved antenna legs, the FIT solver generally underestimated pSAR_10g_. This was due to the limitations of the FIT solver’s hexahedral mesh, which struggled to resolve intricate geometric features like the curved legs of the SN. This insufficient meshing led to errors in calculating port impedance and lumped element properties, which remained unchanged across mesh sweeps for SN in the FIT solver.

The remaining differences in |*B*_1_^+^| field and for pSAR_10g_ between the tools are based on the different meshing techniques (tetrahedral and hexahedral) and meshing implementations, which resulted in inevitable differences in port impedance and the required lumped element values. As noted in the mesh analysis, meshing techniques influence critical parameters such as port impedance, lumped element properties of the tuning and matching networks, and the mass-averaging method for SAR calculations. This emphasizes the need for a standardized methodology in which the lumped elements are explicitly reported to ensure reproducibility. Differences in the tuning and matching network due to software tool constraints resulted in further field variations. For instance, the MS and SN Tx elements were tuned and matched differently in Sim4Life compared to CST and HFSS. Consequently, variations in loss estimation and field results contributed to over- or underestimation of both |*B*_1_^+^| and pSAR_10g_. To compensate for these effects, the SAR efficiency metric (|*B*_1_^+^|/√pSAR_10g_) was proposed as a more reliable measure for reproducibility. The SAR efficiency showed an SD of below 10%, even for complex designs like the MS and SN, which had an SD of <5%. This surprising consistency is largely due to the fine local meshing of the antenna structures, with mesh lines extending into the phantom. The finer phantom meshing enabled more accurate and reproducible results. Two additional metrics—coupling and loading conditions—were also evaluated but demonstrated higher SDs, indicating limited robustness for reproducibility. While these SD values seem concerning, the absolute variations remain relatively minor. For instance, for the SN, the coupling metric exhibited a minimum value of 2.95% and a maximum of 4.68% (SD: 18%). For the MS, the coupling metric ranged from 1.07% to 1.78% (SD: 25%).

The proposed methodology was validated using 7 T MRI experiments, demonstrating comparable agreement for the |*B*_1_^+^| fields between simulation and measurement. The level of agreement was achieved by accounting for signal chain losses in the MRI system, the measured *S*-matrix, and estimated material losses. However, achieving perfect agreement between simulations and experiments remains challenging due to the inherent complexity of MRI experiments as well as uncertainties in material properties and fabrication-related imperfections. In this work, only one |*B*_1_^+^| mapping method was used. Therefore, the measurement uncertainties only rely on the reported inaccuracies of the used AFI sequence [45]. Using other MRI-based |*B*_1_^+^| mapping sequences or field mapping devices could provide additional information and reduce the remaining errors. However, this is out of the scope of this work. Nonetheless, further research towards more harmonized Tx element development must include multiple |*B*_1_^+^| mapping methods as well as cross-site comparisons of the different methods. This study highlights the critical role of a standardized methodology in ensuring reproducible RF Tx element simulations. While certain metrics, such as coupling and loading conditions, exhibited variability, the SAR efficiency metric emerged as a highly robust measure. Using fine meshing techniques for antenna structures and phantom interactions significantly contributed to achieving accurate and reproducible results. These findings advocate for adopting this methodology to support consistent performance evaluations of Tx elements, ultimately advancing research quality and safety standards in UHF-MRI. It is important to note that the proposed methodology is not a rigid framework but rather an exemplary approach to achieving reproducible results across different simulation tools and sites. The reported values demonstrate the feasibility of reproducibility, even when accounting for variations in meshing techniques—such as hexahedral and tetrahedral meshes—and differences in loss estimation, tuning, and matching networks. This adaptability makes the methodology applicable to various single Tx element applications.

Future research could expand on this work by applying the methodology to multi-channel Tx arrays and heterogeneous models. Adopting a standardized methodology applying the FAIR principles has additional benefits, particularly for the scientific review process. It simplifies manuscript evaluation for authors and reviewers by providing clear benchmarks, making it easier to identify novelty and performance improvements. This approach alleviates concerns and critiques often encountered during review processes and streamlines the assessment of work submitted for publication. Furthermore, this methodology can serve as a valuable guide for newcomers to the RF coil research field. Offering a clear framework for evaluating reports and navigating complex topics helps new researchers discern the significance of specific studies and chart a more informed path forward in this challenging area. Such practices not only enhance the reproducibility and quality of scientific research but also foster collaboration, innovation, and progress in the field.

## 5. Conclusions

This study emphasizes the importance of a standardized methodology for the numerical simulation and evaluation of RF Tx elements in UHF-MRI. Variations in simulation tools, modeling assumptions, and meshing techniques have historically confounded reproducibility. Our proposed approach follows the FAIR principles and offers exemplary simulation setups and performance metrics, ensuring consistency and reproducibility across different sites and tools. Our reproducibility study demonstrates its potential as a reliable and straightforward approach for performance assessment, highlighting the significance of fine meshing techniques and localized mesh refinements in improving accuracy and reproducibility for both simple and complex Tx element designs. This methodology is flexible, applicable to a variety of Tx element configurations and simulation tools, and can be extended beyond single Tx elements to multi-channel RF arrays and heterogeneous models. Standardizing simulation practices simplifies the review process, facilitating clearer evaluation of novel contributions. Additionally, this approach provides guidance for newcomers to RF coil research, helping them navigate the complexities of the field. By making the workflow openly accessible following the FAIR principles, we encourage collaboration, transparency, and technology transfer within the MRI community. We urge the adoption of detailed and standardized methodologies for RF Tx element simulations to enhance the reliability, quality, and impact of scientific research in UHF-MRI, driving both technological innovation and patient safety.

## Figures and Tables

**Figure 1 sensors-25-01867-f001:**
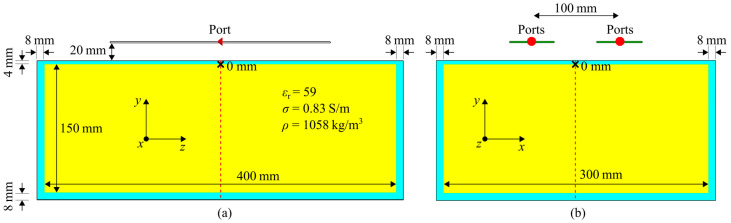
Simulation setup of a single transmit element on the rectangular phantom (*ε*_r_ = 59, *σ* = 0.83 S/m, *ρ* = 1058 kg/m^3^ at 297.2 MHz) with the dimension 400 × 300 × 150 mm^3^ in the central (**a**) sagittal and (**b**) axial plane. The housing of the phantom is shown in blue and the solution in yellow. (**b**) Setup for intrinsic decoupling evaluations with 100 mm center-to-center distance of the antenna. The red dashed line depicts the center axis of the phantom and the black crosses the points of evaluation at 0 mm.

**Figure 2 sensors-25-01867-f002:**
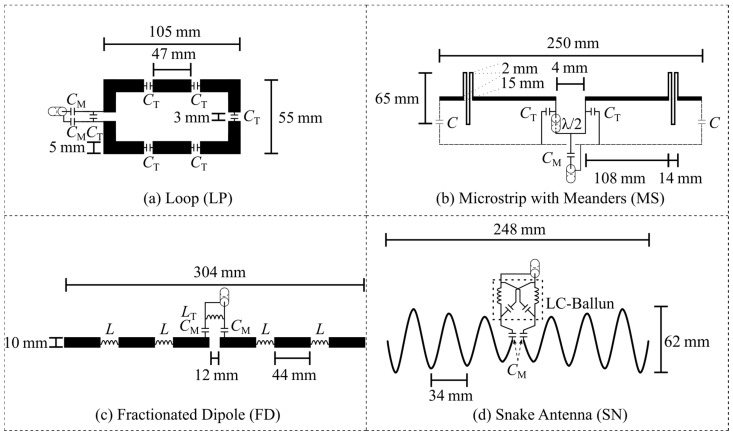
Transmit elements sketched with corresponding dimensions as well as tuning and matching circuits used for the element comparison. Top: (**a**) loop (LP) and (**b**) microstrip with meanders (MS) [28]; Bottom: (**c**) fractionated dipole (FD) [30] and (**d**) snake antenna (SN) [32].

**Figure 3 sensors-25-01867-f003:**
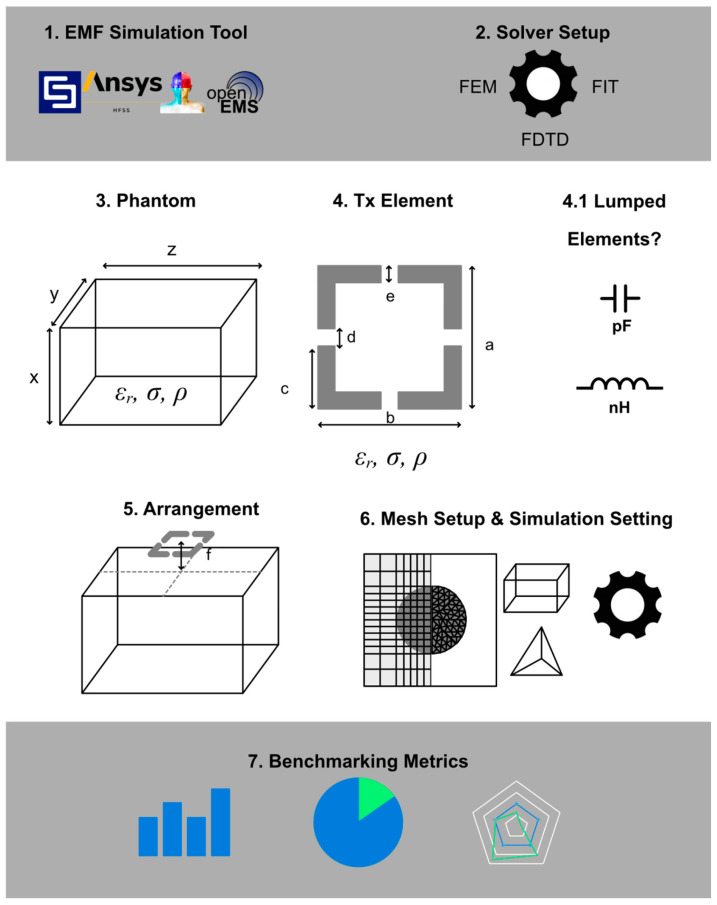
The summarized workflow for simulating radiofrequency Tx elements tailored for UHF-MRI.

**Figure 4 sensors-25-01867-f004:**
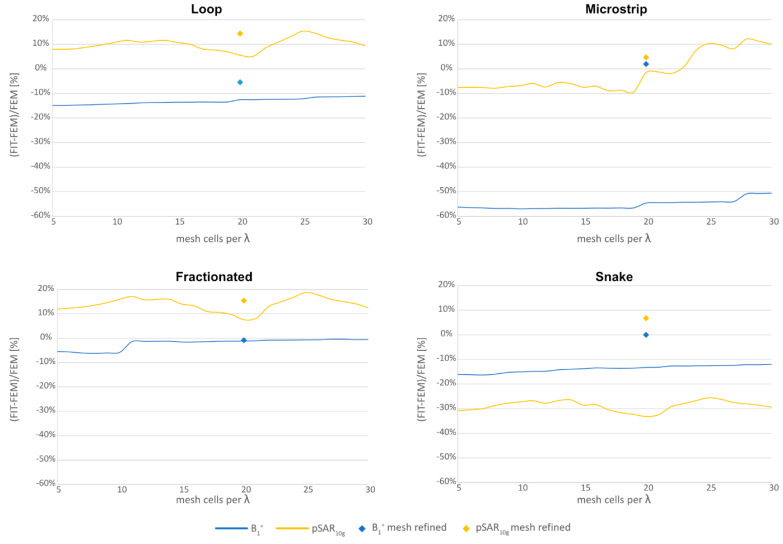
Mesh sweep of the maximum mesh cell size (mesh cells per *λ*) for the loop, fractionated dipole, microstrip with meander, and snake antenna. The relative power efficiency (|*B*_1_^+^|/√*P*) at 0 mm (blue) along the center axis in the phantom, as well as the relative peak SAR10g (yellow) of the FIT solver relative to the FEM solver, is shown. The Tx elements were tuned and matched to |*S*_11_| < −50 dB for each iteration. Higher mesh cells per *λ* refer to finer meshing. The FIT solver results for using the described mesh setting in the EMF simulation setup with local mesh refinements are shown as diamonds.

**Figure 5 sensors-25-01867-f005:**
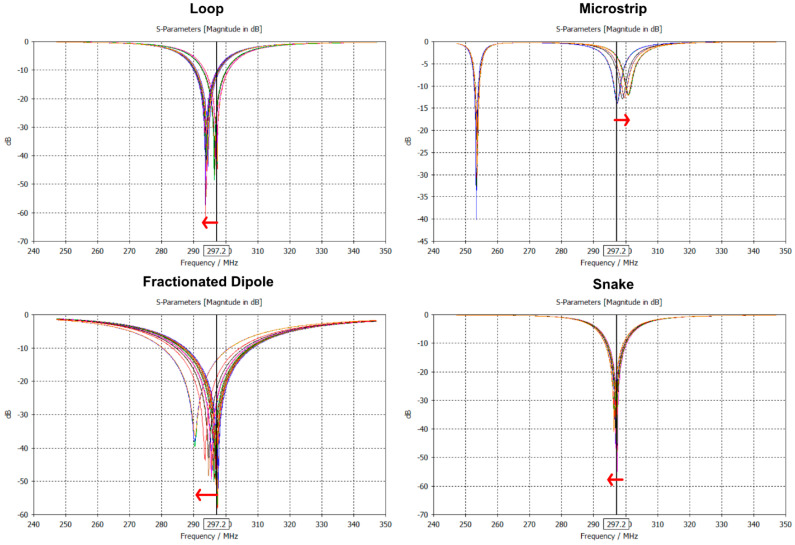
Reflection coefficient (|*S*_11_|) for the loop, fractionated dipole, microstrip with meanders, and snake antenna of the FIT solver with fixed tuning and matching networks for varying maximum mesh cell size (mesh cells per *λ*). The red arrow indicates the frequency shift from 30 to 5 mesh cells per wavelength.

**Figure 6 sensors-25-01867-f006:**
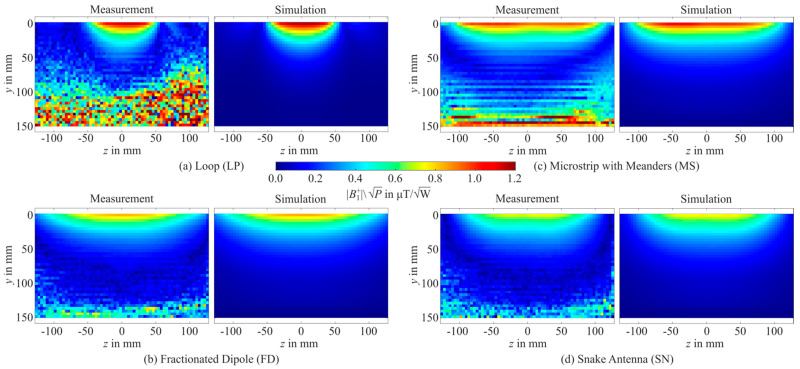
Measured (**left**) and simulated (**right**) axial |*B*_1_^+^|-maps in the center of the phantom for the loop (**a**), the fractionated dipole (**b**), microstrip with meanders (**c**), and the snake antenna (**d**).

**Table 1 sensors-25-01867-t001:** Survey of performance assessment of transmit elements, microstrip with meanders (MS), passively fed dipole (PF), loop (LP), fractionated dipole (FD), self-grounded bow-tie antenna (BT), leaky wave antenna (LW), snake antenna (SN), and coaxial monopole antenna (MP).

	Tx Element	MS [28]	PF [29]	LP [14] ***	FD [30]	BT [16]	LW [31]	SN [32]	MP [33]
Benchmark									
Phantom permittivity *ε*_r_		45.3	50	34	34	48	34	34	46
Phantom conductivity *σ* in S/m		0.87	0.6	0.4	0.4	0.47	0.45	0.4	0.5
Distance Tx element to phantom		30	20	20	20	-	15	20	25
Reference incident power *P*_IN_ in W		0.5	1	1	1	1000	1	1	1
Operating frequency in MHz		297	-	298	298	297.2	300	-	298
|*B*_1_^+^|/√*P* in µT/√W at a given depth or region of interest		**	0.18 *	0.20 *	0.2–0.25 *	4.30_ROI_	0.26_70mm_	0.19 *	0.62_50mm_
Max. pSAR_10g_ in W/kg		0.92	0.98	3.0	1.2 *	-	1.42	1.10	1.10

* Estimated values from line graphs at 100 mm depth; ** only H field evaluated; *** |*B*_1_^+^| for best performing loop and worst case pSAR_10g_ depending on size.

**Table 2 sensors-25-01867-t002:** Mesh analysis results of the loop using (i) adaptive mesh refinement of the FEM solver, (ii) 30 mesh cells per λ of the FIT solver, (iii) 5 mesh cells per λ of the FIT solver, (iv) 20 mesh cells per λ with local mesh refinement of the FIT solver.

Loop	|*B*_1_^+^|_0mm_ in µT/√W	pSAR_10g_ in W/kg	# Mesh Cells	Time in s	Lumped Elements
(i) FEM	1.84	1.60	22,976	73	*C*_T_ = 8.2 pF, *C*_M_ = 16.6 pF
(ii) 30 mesh cells	1.64	1.75	977,730	273	*C*_T_ = 9.7 pF, *C*_M_ = 20.3 pF
(iii) 5 mesh cells	1.57	1.73	74,250	102	*C*_T_ = 9.5 pF, *C*_M_ = 20.6 pF
(iv) Local mesh	1.74	1.83	1,044,630	40	*C*_T_ = 8.4 pF, *C*_M_ = 16.8 pF

**Table 3 sensors-25-01867-t003:** Mesh analysis results of the fractionated dipole using (i) adaptive mesh refinement of the FEM solver, (ii) 30 mesh cells per λ of the FIT solver, (iii) 5 mesh cells per λ of the FIT solver, (iv) 20 mesh cells per λ with local mesh refinement of the FIT solver.

Fractionated	|*B*_1_^+^|_0mm_ in µT/√W	pSAR_10g_ in W/kg	# Mesh Cells	Time in s	Lumped Elements
(i) FEM	1.24	1.23	40,014	139	*L*_T_ = 47.9 nH, *C*_M_ = 42.1 pF
(ii) 30 mesh cells	1.23	1.38	973,700	178	*L*_T_ = 41.2 nH, *C*_M_ = 74.9 pF
(iii) 5 mesh cells	1.17	1.38	61,050	40	*L*_T_ = 36.6 nH, *C*_M_ = 43.5 pF
(iv) Local mesh	1.23	1.42	912,384	172	*L*_T_ = 45.9 nH, *C*_M_ = 45.4 pF

**Table 4 sensors-25-01867-t004:** Mesh analysis results of the microstrip with meanders using (i) adaptive mesh refinement of the FEM solver, (ii) 30 mesh cells per λ of the FIT solver, (iii) 5 mesh cells per λ of the FIT solver, (iv) 20 mesh cells per λ with local mesh refinement of the FIT solver.

Microstrip	|*B*_1_^+^|_0mm_ in µT/√W	pSAR_10g_ in W/kg	# Mesh Cells	Time in s	Lumped Elements
(i) FEM	1.43	1.76	64,605	184	*C*_T_ = 4.8 pF, *C*_M_ = 5.5 pF
(ii) 30 mesh cells	0.71	1.94	1,137,136	227	*C*_T_ = 73.4 pF, *C*_M_ = 302.8 pF
(iii) 5 mesh cells	0.63	1.63	141,120	20	*C*_T_ = 92.7 pF, *C*_M_ = 537.2 pF
(iv) Local mesh	1.46	1.84	2,483,976	575	*C*_T_ = 5.0 pF, *C*_M_ = 5.6 pF

**Table 5 sensors-25-01867-t005:** Mesh analysis results of the snake antenna using (i) adaptive mesh refinement of the FEM solver, (ii) 30 mesh cells per λ of the FIT solver, (iii) 5 mesh cells per λ of the FIT solver, (iv) 20 mesh cells per λ with local mesh refinement of the FIT solver.

Snake	|*B*_1_^+^|_0mm_ in µT/√W	pSAR_10g_ in W/kg	# Mesh Cells	Time in s	Lumped Elements
(i) FEM	1.06	1.18	86,872	173	*C*_M_ = 1.5 pF, *C*_balun_ = 0.05 pF, *L*_balun_ = 116.2 nH
(ii) 30 mesh cells	0.93	0.83	954,720	151	*C*_M_ = 0.7 pF, *C*_balun_ = 0.01 pF, *L*_balun_ = 394.4 nH
(iii) 5 mesh cells	0.89	0.82	61,560	41	*C*_M_ = 0.7 pF, *C*_balun_ = 0.01 pF, *L*_balun_ = 390.7 nH
(iv) Local mesh	1.06	1.26	3,639,944	693	*C*_M_ = 1.3 pF, *C*_balun_ = 0.05 pF, *L*_balun_ = 133.3 nH

**Table 6 sensors-25-01867-t006:** Loop: |*B*_1_^+^| at 0 mm depth, max. pSAR_10g_, SAR efficiency, coupling, loading, and lumped elements with losses calculated with different simulation tools at different sites.

Simulation Tool	|*B_1_*^+^|_0mm_ in µT/√W	pSAR_10g_ in W/kg	SAR Efficiency in µT/√W/kg	Coupling |*S*_21_| in %	Loading |*S*_11_| in %/Δ*f* in MHz	Lumped Elements *C* (ESR), *L*(*Q*)
CST 2020 (FEM)	1.84	1.60	1.45	6.46	0.93/0.9	*C*_T_ = 8.2 pF (0.08), *C*_M_ = 16.6 pF (0.06)
CST 2020 (FIT)	1.74	1.83	1.29	6.76	1.10/1.0	*C*_T_ = 8.4 pF (0.08), *C*_M_ = 16.8 pF (0.06)
CST 2022 (FIT)	1.75	1.85	1.29	6.61	1.12/1.0	*C*_T_ = 8.5 pF (0.08), *C*_M_ = 17.4 pF (0.05)
Sim4Life (FDTD)	1.84	1.89	1.34	6.46	1.05/1.0	*C*_T_ = 8.6 pF, *C*_M_ = 17.1 pF
HFSS (FEM)	1.81	1.76	1.36	5.01	1.32/1.2	*C*_T_ = 8.3 pF (0.27), *C*_M_ = 21.1 pF (0.27)
Mean	1.79	1.79	1.34	6.26	1.10/1.02	
SD	0.06	0.10	0.07	0.71	0.14/0.11	

**Table 7 sensors-25-01867-t007:** Fractionated dipole: |*B*_1_^+^| at 0 mm depth, max. pSAR_10g_, SAR efficiency, coupling, loading, and lumped elements with losses calculated with different simulation tools at different sites.

Simulation Tool	|*B*_1_^+^|_0mm_ in µT/√W	pSAR_10g_ in W/kg	SAR Efficiency in µT/√(W/kg)	Coupling |*S*_21_| in %	Loading |*S*_11_| in %/Δ*f* in MHz	Lumped Elements *C* (ESR), *L* (*Q*)
CST 2020 (FEM)	1.24	1.23	1.12	2.24	0.38/1.3	*L* = 33.5 nH (55.0), *L*_T_ = 47.9 nH (55.0), *C*_M_ = 42.1 pF (0.03)
CST 2020 (FIT)	1.23	1.42	1.03	2.40	0.42/1.4	*L* = 33.5 nH (55.0), *L*_T_ = 45.9 nH (55.0), *C*_M_ = 45.4 pF (0.03)
CST 2022 (FIT)	1.24	1.44	1.03	2.40	0.79/2.4	*L* = 33.5 nH (55.6), *L*_T_ = 48.5 nH (59.8), *C*_M_ = 38.0 pF (0.04)
Sim4Life (FDTD)	1.22	1.41	1.03	2.45	0.42/1.3	*L* = 33.5 nH (55.6), *L*_T_= 58.7 nH (55.6), *C*_M_= 46.2 pF
HFSS (FEM)	1.30	1.56	1.04	2.69	0.46/1.5	*L* = 33.5 nH (230), *L*_T_ = 13.4 nH (230), *C*_M_ = 6.9 pF (0.27)
Mean	1.25	1.41	1.05	2.44	0.49/1.58	
SD	0.03	0.11	0.04	0.16	0.17/0.47	

**Table 8 sensors-25-01867-t008:** Microstrip: |*B*_1_^+^| at 0 mm depth, max. pSAR_10g_, SAR efficiency, coupling, loading, and lumped elements with losses calculated with different simulation tools at different sites.

Simulation Tool	|*B*_1_^+^|_0mm_ in µT/√W	pSAR_10g_ in W/kg	SAR Efficiency in µT/√W/kg	Coupling in |*S*_21_| in %	Loading |*S*_11_| in %/Δ*f* in MHz	Lumped Elements *C* (ESR), *L* (*Q*)
CST 2020 (FEM)	1.43	1.76	1.08	1.07	0.17/0.1	*C*_T_ = 4.8 pF (0.05), *C*_M_ = 5.5 pF (0.05), *C*_end_ = 1 pF (0.12)
CST 2020 (FIT)	1.46	1.84	1.08	1.78	0.35/0.4	*C*_T_ = 5.0 pF (0.05), *C*_M_ = 5.6 pF (0.05), *C*_end_ = 1 pF (0.12)
CST 2022 (FIT)	1.46	1.86	1.07	1.10	0.43/0.4	*C*_T_ = 5.0 pF (0.09), *C*_M_ = 5.6 pF (0.09), *C*_end_ = 1 pF (0.16)
Sim4Life (FDTD)	1.51	1.94	1.08	1.07	0.19/0.3	*C*_T_ = 9.2 pF, *C*_M_ = 3.9 pF, *C*_end_ = 1 pF (0.12)
HFSS (FEM)	1.41	1.79	1.05	1.12	0.23/0.3	*C*_T_ = 10.6 pF (0.27), *C*_M_ = 6.8 pF (0.27), *C*_end_ = 1 pF (0.27)
Mean	1.45	1.84	1.07	1.23	0.27/0.3	
SD	0.04	0.07	0.01	0.31	0.11/0.1	

**Table 9 sensors-25-01867-t009:** Snake antenna: |*B*_1_^+^| at 0 mm depth, max. pSAR_10g_, SAR efficiency, coupling, loading, and lumped elements with losses calculated with different simulation tools at different sites.

Simulation Tool	|*B*_1_^+^|_0mm_ in µT/√W	pSAR_10g_ in W/kg	SAR Efficiency in µT/√W/kg	Coupling |*S*_21_| in %	Loading |*S*_11_| in %/Δ*f* in MHz	Lumped Elements *C* (ESR), *L* (*Q*)
CST 2020 (FEM)	1.15	1.25	1.03	2.95	0.30/0.3	*C*_balun_ = 9.35 pF (0.04), *L*_balun_ = 21.43 nH (49.7), *C*_M_ = 4.18 pF (0.05)
CST 2020 (FIT)	1.15	1.47	0.95	3.80	0.49/0.7	*C*_balun_ = 10.05 pF (0.04), *L*_balun_ = 23.90 nH (49.7), *C*_M_ = 3.66 pF (0.06)
CST 2022 (FIT)	1.15	1.48	0.95	3.80	0.48/0.7	*C*_balun_ = 11.16 pF (0.04), *L*_balun_ = 28.29 nH (49.7), *C*_M_ = 3.83 pF (0.06)
Sim4Life (FDTD)	1.20	1.56	0.96	4.68	0.51/0.6	*C*_S_ = 2.1 pF, *C*_P_ = 0.1 pF
HFSS (FEM)	1.16	1.52	0.94	3.24	0.44/0.6	*C*_balun_ = 11.05 pF (0.27), *L*_balun_ = 25.36 nH (230), *C*_M_ = 4.17 pF (0.27)
Mean	1.16	1.46	0.97	3.67	0.46/0.6	
SD	0.02	0.12	0.04	0.66	0.05/0.2	

## Data Availability

The antenna models with the phantom are available under https://github.com/AntennaComp (accessed on on 1 November 2024). The standardized protocol is available at https://www.opensourceimaging.org/transmit-element-comparison/ (accessed on on 1 November 2024). The data presented in this study are available on a reasonable request from the corresponding author.

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
