# Peer review of "Reproducibility of Electromagnetic Field Simulations of Local Radiofrequency Transmit Elements Tailored for 7 T MRI"

_sensors, 2025, doi:10.3390/s25061867_

Round 1
Reviewer 1 Report
Comments and Suggestions for Authors
This manuscript provides invaluable resources and guidelines for reproducible research in ultra-high field MRI transmit (Tx) RF coil development, including well-defined simulation methodologies and performance metrics. With the help of these guidelines, the coil simulation reproducibility was demonstrated under varying test conditions (by using five simulations solvers) and testers (three imaging sites). The developed guideline is available open-source online and promotes technology transfer, research reproducibility, and lowers barriers to the accessibility of the state-of-the-art engineering. I recommend this work for publication as it provides a useful and reliable tool for MRI RF engineers. As is, the manuscript provides clear guidelines for experienced RF engineers, however additional software-specific settings demonstrated as screenshots of the actual simulated model would enhance this work and provide a helpful tutorial-like resource of younger engineers who just start learning about the UHF -MRI RF coil simulation.
Mesh Analysis: It would be beneficial to show the mesh analysis examples for all software tools tested, because, as the authors mentioned, the meshing techniques and analysis strategies vary between the software tools and the given CST-specific meshing analysis steps may not directly translate to other software tools. It would also be very helpful to provide screenshots of the meshing parameters for each software as well as the meshed model itself (maybe highlighting just the critical parts that require at least two mesh cells like the ports/traces). These screenshots don’t have to be in the main body of the manuscript and can go to supplementary files instead, but they could definitely enhance the value of the work, especially for young researchers just starting to use the tools and learning how to properly model RF coils.
Line 265: tuning and matching of the Tx coil is given as part of the mesh analysis, however, it can be a separate step in itself. It would be helpful to provide some guidelines on this step as well, as it varies depending on the software. For example, CST allows to simulate all lumped elements as ports and then find the perfect tune/match parameters in the schematic; however, in Sim4Life this step must be done manually (if losses are included). Providing screenshots and describing the setting differences between the software tools would be of tremendous help.
Other simulation settings: it would be helpful to provide other general and software-specific simulations settings, for example, how big of a (air) domain to use (relative to the phantom size of wavelength)? What are the boundary conditions? (software-specific) What is the trace line thickness (or its infinitesimally small)? Etc.
Reproducibility, line 280: it would be good to provide the exact instructions given to the testers (supplementary material)
Tables 2-5: wouldn’t it be more valuable to see in one table the result comparison of different meshing settings specific to a particular coil geometry? Meaning, Tables 2: Title: Results for the LP coil element; Rows: different mesh settings (adaptive, 5, 20, 30 cells); Columns: (same as in the original tables). This way, a reader can directly compare the results under different meshing settings.
It's not clear what mesh setting were used in the following reproducibility tests for each coil geometry and software tool.
Results: Reproducibility: The alternative matching circuits need to be shown for the MS and SN coils that could not be tuned and matched with the prescribed method (CST) in Sim4Life and HFSS. Again, this is where the authors could potentially add more value to the manuscript by highlighting the differences between the software tools and recommending the work-around strategies with schematics and screenshots.
How was the coupling simulated? Please show the model setups for each coil geometry and S21 results.
Reviewer 2 Report
Comments and Suggestions for Authors
Comments to the Authors
This study aims to evaluate the feasibility of a standardized methodology for electromagnetic simulations to achieve reproducible RF transmit coils. The simulations were conducted using CST and validated under a 7T MRI scanner with B1 mapping techniques. Reproducibility was assessed by comparing CST results with other simulation platforms such as Sim4Life and HFSS. The study provides a useful guideline for individuals new to electromagnetic simulations. However, it lacks significant novelty and innovative contributions, making it unsuitable as a research paper.
Introduction:
- L53: SNR or sensitivity? Please clarify the distinction.
- L56: Maintain consistency in referring to UHF-MRI.
- L63: Consider reviewing the monopole Tx configuration. A more comprehensive literature review is needed.
- L108: Should this reference a table?
- L110 (Table 1): Modify the table to include the monopole structure. What is the operating frequency for each configuration?
Methods:
- L136: ± 50 MHz is a large margin that may significantly impact wavelength effects.
- L151: Include a 3D schematic of the phantom. Explain the rationale behind its specific dimensions. Please perform the simulations using the human model as it may provide more realistic results regarding the validation.
- L158: Use "Tx coils" instead of "Tx elements" for consistency.
Results:
- L315: Define λ before mentioning it.
- L357: The matching techniques in CST and Sim4Life differ. Please elaborate on this.
- L389: Do you mean Δ = 6.2 MHz? Please clarify.
- L394: In Sim4Life, simulations can be performed with the same phase, and a λ/2 phase difference can be applied in post-processing.
Discussion:
- L545: Additionally, discuss:
- The effect of different permittivity and conductivity values on simulation results.
- How does the distance between the coil element and the phantom affect the performance of the coil element, including B1 and SAR efficiency?
Reviewer 3 Report
Comments and Suggestions for Authors
Greetings!
I really liked your article!
See the review in the attached (.doc) file.

Round 2
Reviewer 2 Report
Comments and Suggestions for Authors
I appreciate the authors' efforts in addressing most of the concerns raised in the initial review. Their revisions have significantly improved the clarity and robustness of the manuscript. However, Comment 8 regarding simulations using a human model remains unaddressed. Incorporating such simulations could provide more realistic results, particularly in terms of validation. I encourage the authors to consider this aspect to further strengthen their study.
Author Response
Comments and Suggestions for Authors
I appreciate the authors' efforts in addressing most of the concerns raised in the initial review. Their revisions have significantly improved the clarity and robustness of the manuscript. However, Comment 8 regarding simulations using a human model remains unaddressed. Incorporating such simulations could provide more realistic results, particularly in terms of validation. I encourage the authors to consider this aspect to further strengthen their study.
Response:
Thank you for recognizing our efforts to improve the manuscript. While human voxel models yield more realistic results, especially for Tx element safety assessments, reproducibility studies with such human models are hardly feasible due to the lack of cross-platform human models. For instance, HFSS poses an internal human mesh model with tetrahedral meshing, while CST meshes the imported human voxel model with hexahedral mesh cells. Thus, a comparable meshing of the human models can not be guarantee,d especially in the transition regions between organs.
Validating human models will be very challenging because human voxel models only mimic average human bodies with fixed geometry and electric parameters, which do not fully reflect real human variability in torso geometry, permittivity, and conductivity. Building such an anthropomorphic human phantom would be nearly impossible, and benchmarking against real humans adds additional error sources and likely results in greater discrepancies between simulation and measurement than reported in this study.
Simple-shaped phantoms, as used here, allow better alignment between simulation and experiment by minimizing model differences and placement errors. This is the reason most literature uses uniform phantoms.
Our aim is not to compare Tx element types but to demonstrate reproducibility among users, sites, and simulation software platforms. Developing comprehensive guidelines for human voxel models, including the standard human model type and the region of interest, lies beyond this manuscript's scope.
Based on these considerations, we believe human voxel model simulations would not significantly enhance the quality of the manuscript and are therefore not added.